# Cystic Fibrosis Human Organs-on-a-Chip

**DOI:** 10.3390/mi12070747

**Published:** 2021-06-25

**Authors:** Herbert Luke Ogden, Hoyeol Kim, Kathryn A. Wikenheiser-Brokamp, Anjaparavanda P. Naren, Kyu Shik Mun

**Affiliations:** 1Division of Pulmonary Medicine, Cystic Fibrosis Research Center, Department of Pediatrics, Cincinnati Children’s Hospital Medical Center, Cincinnati, OH 45229, USA; ogdenhl@mail.uc.edu; 2Division of Pulmonary and Critical Care Medicine, Department of Medicine, Cedars-Sinai Medical Center, Los Angeles, CA 90048, USA; Hoyeol.Kim@cshs.org; 3Division of Pathology and Laboratory Medicine, The Perinatal Institute, Cincinnati Children’s Hospital Medical Center, Cincinnati, OH 45229, USA; Kathryn.Wikenheiser-Brokamp@cchmc.org; 4Division of Pulmonary Biology, Cincinnati Children’s Hospital Medical Center, Cincinnati, OH 45229, USA; 5Department of Pathology and Laboratory Medicine, University of Cincinnati College of Medicine, Cincinnati, OH 45219, USA; 6Board of Governors Regenerative Medicine Institute, Department of Pediatrics, Cedars-Sinai Medical Center, Los Angeles, CA 90048, USA

**Keywords:** organ-on-a-chip, cystic fibrosis, CFTR, personalized medicine

## Abstract

Cystic fibrosis (CF) is an autosomal recessive disease caused by mutations in the cystic fibrosis transmembrane regulator (CFTR) gene: the gene product responsible for transporting chloride and bicarbonate ions through the apical membrane of most epithelial cells. Major clinical features of CF include respiratory failure, pancreatic exocrine insufficiency, and intestinal disease. Many CF animal models have been generated, but some models fail to fully capture the phenotypic manifestations of human CF disease. Other models that better capture the key characteristics of the human CF phenotype are cost prohibitive or require special care to maintain. Important differences have been reported between the pathophysiology seen in human CF patients and in animal models. These limitations present significant limitations to translational research. This review outlines the study of CF using patient-derived organs-on-a-chip to overcome some of these limitations. Recently developed microfluidic-based organs-on-a-chip provide a human experimental model that allows researchers to manipulate environmental factors and mimic in vivo conditions. These chips may be scaled to support pharmaceutical studies and may also be used to study organ systems and human disease. The use of these chips in CF discovery science enables researchers to avoid the barriers inherent in animal models and promote the advancement of personalized medicine.

## 1. CFTR Mutations

Since the first description of CF in 1938 [1], over 2000 mutations in the CFTR gene that produce a broad range of patient phenotypes have been discovered [2,3]. Presently, mutations are categorized into six classes based on their primary biological defect (Figure 1) [4]. **Class I** (synthesis) mutations result in a total or partial loss of CFTR protein expression caused by the introduction of a premature termination codon [3]. This loss of CFTR protein expression leads to a complete loss of CFTR chloride channel function in the affected epithelia. Class I mutations are present in ~10% of patients with CF. Common examples of class I mutations include G542X, W1282X, and R552X. **Class II** (misfolding) mutations are associated with defective processing due to a misfolded CFTR protein [2]. During processing, the misfolded protein is targeted for degradation through the ubiquitin pathway. Class II mutations are the most common mutations in the CF population, with over 85% of patients having the F508del mutation [4,5]. **Class III** (regulation) mutations are gating mutations found in the ATP binding domains of the CFTR protein. A missense mutation in this gene region produces a protein that is unable to be phosphorylated, which causes a decrease in chloride channel activity [6]. The most common mutation in this category is G551D, and other, less common mutations include V530F and S549R. **Class IV** (conduction) mutations are associated with alterations in the membrane spanning domain of the CFTR. These missense mutations produce a protein that has reduced cAMP-dependent chloride channel activity [2]. Individuals with class IV mutations usually present with milder CF symptoms. Class IV mutations are detected in less than 3% of CF patients, with the most common examples including R117H, D1152H, and R347P. **Class V** (mRNA expression) mutations result in a total reduction in the CFTR protein, usually through pre-mRNA splicing. These splicing errors result in the decreased expression of CFTR on the cell surface. Class V mutations are detected in less than 3% of CF patients, with the most common mutations including A455E [7], 3849 + 10kbC > T [8], 2789 + 5G > A [8], and 3272 − 26A > G [9]. **Class VI** (CFTR instability) mutations, including the Q1412X mutation, cause the instability of a functional CFTR protein [10]. This instability causes increased CFTR turnover on the apical surface of cells.

CF animal models (e.g., mouse [11,12,13], ferret [14,15], pig [16], rat [17], monkey [18], and sheep [19,20]) have been generated to recapitulate human CF mutations, but it is challenging to reproduce human disease with animal models (e.g., CF mice have normal lung function) [21]. To develop a novel in vitro CF model that helps researchers reproduce human organ physiology to investigate patient-specific disease is critical.

## 2. Rescue of CFTR Function

### 2.1. Potentiator Treatment

Until recently, medical treatment of CF focused on targeting the organ-specific sequelae from the underlying disease [22,23]. However, multiple treatments that target mutation-specific defects of the CFTR protein have recently been discovered. The first drug to receive FDA approval was ivacaftor (Kalydeco, Vertex Pharmaceuticals, Boston, MA, USA), a CFTR “potentiator” that works by opening the dysfunctional CFTR channel present on the cell surface [24,25,26,27,28,29]. A CFTR potentiator acts on the CFTR gating mutations and increases the probability of the CFTR channel opening. Improved opening of the CFTR channel enhances chloride and bicarbonate secretion, leading to sodium and water flow caused by the osmotic driving force that helps in maintaining the pH and water level on the apical surface. Two key phase III clinical studies showed that treated patients with at least one copy of the G551D CFTR gating mutation had marked improvement in lung function (forced expiratory volume in 1 s (FEV_1_)), quality of life, weight, and biomarkers of CFTR function (sweat chloride) [25,28]. As a result of these studies, the FDA approved the use of ivacaftor in 2012 for the treatment of patients with the G551D mutation who were aged 6 years of age and older. Based on additional clinical studies in individuals with other non-G551D class III gating mutations, ivacaftor is now approved for patients with any one of eight additional gating mutations (G1244E, G1349D, G178R, G551S, S1251N, S1255P, S549N, or S549R) [30,31]. Although the G551D mutation is only present in 5% of the CF population, the success of ivacaftor showed that modulators are effective in rescuing CFTR protein function, opening the door for the discovery of additional novel modulators that may be relevant to a broader population of CF patients.

### 2.2. Corrector Treatment

While ivacaftor showed improvement in FEV_1_ in patients with G551D CFTR mutations, studies assessing ivacaftor efficacy in patients with the most common F508del CFTR mutation did not produce the same beneficial outcomes. These studies concluded that while ivacaftor remained a safe treatment, its poor efficacy suggested that ivacaftor alone was not an effective therapeutic approach for patients with homozygous F508del CFTR mutations [32]. In another study, treatment with a combination drug, lumacaftor/ivacaftor (Orkambi, Vertex Pharmaceuticals, Boston, MA, USA), showed improved lung function and nutritional status as well as decreased pulmonary exacerbations in CF patients with severe lung disease caused by homozygous F508del CFTR mutations who were 12 years of age or older [33,34]. Lumacaftor acts as a corrector helping to bring the CFTR protein to the membrane surface. Lumacaftor alone does not improve CF patient outcomes and must be taken in combination with a potentiator such as ivacaftor. An effective treatment regimen included both lumacaftor (400 mg) and ivacaftor (250 mg) every 12 h for 24 weeks. The study also noted that in some patients, treatment dosages were decreased to reduce adverse events. The study enrolled a total of 46 patients and showed that lumacaftor/ivacaftor combination treatment was associated with reductions in the duration of IV antibiotics for exacerbation-related symptoms and in the total number of hospitalizations [35].

A second corrector (tezacaftor; vx-661) was approved by the FDA in 2018 as a medication for use in combination with ivacaftor (Symdeko; tezacaftor/ivacaftor; Vertex Pharmaceuticals) in patients 12 years of age or older who have CF and are homozygous for F508del mutations [35,36]. Patients were randomly assigned to receive either 100 mg of tezacaftor once daily and 150 mg of ivacaftor twice daily or a matched placebo for 24 weeks. The study reported that combination treatment with tezacaftor/ivacaftor was efficacious and safe in patients 12 years of age and older who were homozygous for the F508del CFTR mutation [36].

### 2.3. Trikafta Treatment

A new combination tablet composed of elexacaftor (vx-445), tezacaftor, and ivacaftor was FDA approved in 2019 for the treatment of CF patients 6years of age and older who carry at least one F508del mutation (Trikafta, Vertex Pharmaceuticals, Boston, MA, USA) [37,38,39,40]. It is recommended that one tablet containing 100 mg elexacaftor, 50 mg tezacaftor, and one tablet containing only 75 mg ivacaftor be taken in the morning, followed by taking an additional 150 mg of ivacaftor in the evening, approximately 12 h later [39,40]. Gopalkumar Rakesh’s group reported the case of a 19-year- old CF patient diagnosed at aged 4 with F508del/1898 + 1G > A mutations and a sweat chloride level of 105 mmol/. Within two weeks of taking this medication combination, she showed a remarkable improvement in FEV_1_ from 60% to 83%. This case highlights the promising outcomes associated with combination therapy [26].

Although many studies demonstrate that CF patients may respond favorably to combination therapy, responses to CFTR modulator therapies may be variable, and the mechanisms underlying variable therapeutic responses remain a mystery. Limited numbers of patients have CFTR mutations other than F508del and thus, randomized placebo-controlled trials to guide treatment in patients with rarer genotypes are lacking. Consequently, there is a growing need to develop advanced personalized study designs to predict and evaluate individual responses to emerging CFTR therapies [23].

## 3. CF Organs

### 3.1. Pancreas

The terminal portion of the pancreatic duct connects to the acinar cells within the pancreas and plays an important role in delivering powerful digestive enzymes into the duodenum [41,42]. The CFTR protein is highly expressed in the pancreatic ductal epithelial cells, allowing fluid and anions to enter the ductal lumen to maintain the pH balance [43]. Impaired exocrine pancreatic function is a common manifestation of CF, resulting in exocrine pancreatic insufficiency (EPI). CFTR mutations are associated with the impaired pancreatic ductal secretion of anions and fluids, resulting in reduced intraluminal pH and a low secretory volume [44]. Approximately 85% of infants diagnosed with CF have EPI at birth. EPI can cause malabsorption and maldigestion, with common symptoms including diarrhea, flatulence, abdominal pain, and steatorrhea (large volumes of foul-smelling stools) [45]. The 2016 Cystic Fibrosis Patient Registry reports that more than 80% of patients with CF are prescribed pancreatic enzyme replacement therapy (PERT), with proper PERT administration at the appropriate dose relieving many of the CF-related pancreatic symptoms.

Pancreatic islets are composed of multiple cell types and produce hormones that are secreted directly into the blood stream, including insulin (β cell; 60%), glucagon (α cell; 30%), somatostatin (δ cell), and pancreatic polypeptide (pp cell) [46,47,48,49,50,51]. CFTR is only expressed in the pancreatic ductal epithelial cells in the human pancreas, and defective CFTR function indirectly causes impaired endocrine pancreatic function, resulting in cystic fibrosis-related diabetes (CFRD) [52,53]. In general, the powerful digestive enzymes produced by the acinar cells are inactivated in the pancreatic duct (pH ~8.3) [54] and are then activated in the duodenum due to the lower pH (pH ~ 6) [41,54]. However, defective CFTR function in the pancreatic ductal epithelial cells causes a lack of bicarbonate secretion that leads to prematurely activated digestive enzymes in the pancreatic duct lumen. This inappropriate enzyme activity promotes pancreatic tissue damage, leading to pancreatic fibrosis and decreased insulin production by β-cells. Previous studies demonstrate that defective CFTR function in the pancreatic ductal epithelial cells located in close proximity to the pancreatic islets directly reduces insulin secretion from the pancreatic islets [55].

### 3.2. Gastrointestinal Tract

The human gastrointestinal tract harbors a complex and dynamic population of microorganisms [56] that affect digestive, immune, exocrine, and endocrine function [57]. CFTR plays a key role in the regulation of salt and water efflux in the intestinal lumen, which is critical for maintaining both the fluidity and pH of the luminal contents. Defective CFTR function decreases the flux of bicarbonate and chloride, causing increased mucus viscosity and decreased intraluminal fluidity. Chronic inflammation, dysmotility, and dysbiosis [58], referred to as “CF gut”, then develop. The resulting increased acidity can damage the epithelial cell lining in the small intestine and can lead to dysfunctional digestive enzymes. Intestinal dehydration also results in the accumulation of viscous mucus, which can lead to abdominal pain, cramping, gas, greasy stools, flatulence, or intestinal obstruction [58,59,60]. In neonatal CF patients, this obstruction, known as meconium ileus, can result in intestinal rupture and sepsis if left untreated. In adults with CF, distal intestinal obstructive syndrome (DIOS) and constipation may occur. Approximately 30–40% of individuals with CF suffer from small intestinal bacterial overgrowth (SIBO) as a consequence of the increased mucus accumulation in the intestine [58].

### 3.3. Lung

Cystic fibrosis affects the airway surface by altering the airway surface liquid (ASL) covering the luminal surface of the airway epithelium. ASL is a critical contributor to the antimicrobial properties of the airway, functioning to maintain ciliary function and mucociliary clearance [61,62]. Mutations in CFTR prevent chloride and bicarbonate ion diffusion, resulting in decreased water flow and airway surface dehydration as well as the creation of viscous mucus that is difficult to clear. An additional deleterious effect of this viscous mucus layer is ineffective mucociliary transport in the lungs, predisposing patients to chronic infection and inflammation that lead to declined lung function [61]. Decreased pulmonary function is usually monitored by FEV_1_. Continued cycles of infection and inflammation ultimately lead to lung fibrosis and respiratory failure.

While the alveolar-capillary interface remains an important component of pulmonary function, growing evidence suggests that pulmonary endothelium and smooth muscle have critical functions in maintaining low pressure pulmonary circulation and allowing for effective gas exchange. Anatomically, airway smooth muscle (ASM) is peripherally arranged in the airway walls to control luminal diameter and is found in the posterior aspect of the trachea lacking cartilaginous rings. While the anatomic structure of the lung is well known, the precise function of ASM is not clear. Multiple reports emphasize that ASM functions as a dynamic and complex cell and regulatory structure that produces signaling factors that contribute to patient phenotypes; for these reasons, ASM is a potential target for novel therapeutic approaches [63]. Current culture methods exist to grow both pulmonary arterial endothelial cells (PAECs) and microvascular endothelial cells (MVECS) for mechanistic research studies [64]. Successfully harvesting these human PAECs and MVECS allows for the development of tractable models. Such models support mechanistic studies that further enhance understanding of pulmonary vascular disease.

The literature reports conflicting data regarding CFTR expression in distal regions of the human lung. For example, Kreda et al. reported high levels of CFTR expression in small airway epithelium with minimal CFTR expression in the alveoli utilizing in situ hybridization [65]. However, Engelhardt et al. demonstrated detectable levels of CFTR mRNA in both the bronchioles and alveoli [66]. A more recent study by Regnier et al. utilized immunohistochemistry with anti-CFTR antibodies, MAB25031 and MAB1660, to demonstrate CFTR expression in bronchiolar epithelium as well as both type I and II alveolar cells [67]. This report also mentioned that higher CFTR expression may be present in the bronchiolar epithelium, but the precise level of CFTR expression was not measured. Taken together, this combination of findings supports the existence of CF pathophysiology in small airways and underscores the need for further studies on the role of CFTR in type I and II alveolar cells.

While the role of CFTR in epithelial cells is well documented, limited studies address the role of CFTR in ASM. CF pulmonary disease manifestations include airway inflammation, bacterial infection, mucus accumulation, and airflow obstruction. While airflow obstruction is multifactorial, alterations in ASM may contribute to the disease phenotype. There are three main lines of evidence supporting a role of CFTR in ASM that exist. First, many CF patients present with symptoms similar to asthma and airway bronchoconstriction [68,69]. Airway smooth muscle remodeling is a hallmark feature of asthma, and studies show that similar pathophysiology occurs in CF patients with mild to moderate airway obstruction [69]. While inflammation likely contributes to the altered ASM function seen in later stage CF, the role of CFTR in ASM function in earlier stages of CF is not well known.

Second, prior studies have shown the role of CFTR in smooth muscle function [70,71,72]. As an example, structural abnormalities were observed in the airways of CF mice and rats (CFTR^−/−^) [73,74]. However, CFTR^−/−^ mice still secrete chloride and thus demonstrate pathophysiology that differs from human CF [75]. In CFTR^−/−^ pigs that develop airway disease mimicking human disease, structural airway abnormalities are present at birth, suggesting that CFTR may be important in ASM development [70,71]. In the absence of airway inflammation and infection, CFTR^−/−^ pigs display irregularly shaped cartilage and abnormal tracheal smooth muscle [70]. These observations suggest that a defect in CFTR function in the ASM may contribute to the underlying pathophysiology. In vitro studies examining human F508del CFTR mutant ASM cells showed positive CFTR staining and CFTR channel involvement in intracellular Ca^2+^ regulation in response to a contractile agonist [72]. Support from additional studies showed in vitro activation of CFTR in human ASM independent of the bronchial epithelium [76]. A more recent study further supports CFTR function in the ASM by demonstrating that CFTR localizes to the sarcoplasmic reticulum in the ASM and that a loss of CFTR function results in an altered contractile phenotype [77,78]. Finally, treatment with ivacaftor, a CFTR potentiator, showed rapid recovery in pulmonary function, which was evident by an increase in FEV_1_ in CF patients. These observations related to airflow obstruction may be mediated in part by improvement in ASM function. Together, this evidence of CFTR function in the ASM stresses the need to further investigate CFTR function as it relates to ASM-related CF phenotypes.

## 4. Clinical Trials and N-of-1 Studies

While clinical trials remain the gold standard in drug development and testing, this study design poses increasing challenges for CF patients. Current estimates suggest only ~30% of all CF patients are eligible to enroll in clinical studies [79]. The demand for patient enrollment in clinical studies remains high despite the small CF population size. A lack of patient enrollment prevents researchers from meeting the minimum required number of individuals for phase II and phase III clinical trials. Furthermore, both the age of study subjects and a lack of infrastructure may further limit the number of available candidates for clinical trials.

Another concern with regard to clinical studies is the varying degrees of disease severity seen in CF patients who present with wide variability in clinical manifestations, from asymptomatic to severely functionally impaired. This severity spectrum has implications when reporting adverse events in CF patients. Adverse drug effects continue to be a key safety endpoint in CF subjects. However, there remains a lack of published data assessing adverse effects in these patients [80]. A study by Sucharew et al. reports that CF subjects have relatively high rates of adverse events during clinical trials. Surprisingly, this high adverse event rate is seen in both test and placebo subjects in reports combining respiratory adverse event rates over multiple studies [80].

There continues to be ongoing debate as to the effectiveness of the current endpoint that is most commonly used as a clinical indicator of drug efficacy, sustained FEV_1_ [81]. An effective endpoint measurement requires that statistical significance can be reasonably demonstrated while also being clinically relevant. Although statistically significant changes may be identified, FEV_1_ may not be a clinically significant endpoint when assessing the long-term outcomes in CF patients [82]. A possible alternative endpoint would be the mean rate of FEV_1_ decline. This measurement captures the stabilization of lung function over time, providing a more clinically relevant endpoint. However, the mean rate of FEV_1_ decline is more statistically demanding. Using this endpoint would require patients to be monitored for a longer duration to provide accurate results. Using a more complicated endpoint may therefore require patients to be engaged for longer periods of time and may further reduce the already small pool of CF patients who are available and willing to participate in clinical trials.

While ivacaftor and lumacaftor/ivacaftor treatments have shown success, the vast majority of CF mutations are rare, making it difficult to study and treat using conventional clinical trial study designs. The current FDA approval for ivacaftor therapy is limited to approximately 5% of all CF patients [83]. With such a low percentage of the CF population receiving treatment, it raises the question of whether patients with rare CFTR mutations may benefit from treatment but lack access due to restricted FDA approval of the use of modulator therapy. The current standard for clinical trials, the randomized controlled trial (RCT), requires a large homogenous population to predict drug efficacy [84]. Due to the number of known mutations in CFTR, it is impossible to conduct these RCTs on ivacaftor treatment for every possible mutation. The N-of-1 study design can be used to close this gap by conducting rigorous investigations into the treatment effectiveness of CFTR modulators on individual patients with various CFTR mutations [84]. The N-of-1 study concept works to determine the optimal intervention for a single patient, with the end goal of personalized care for each patient [85].

A clinical study conducted at the University of San Francisco implemented a single subject N-of-1 study design to test the effectiveness of ivacaftor in patients with class III CFTR mutations. Subjects were randomized to the oral administration of ivacaftor twice a day for 14 days followed by placebo for 14 days or vice versa [84]. In this manner, each patient served as his or her own control. Sweat chloride concentration was the outcome used to determine if the drug was effective. The results were mixed with some, but not all, individuals showing a reduced sweat chloride concentration with ivacaftor. Two subjects had increased sweat chloride concentrations while on the drug. These results demonstrate that individuals respond differently to treatment and further supports the need for the development of personalized approaches to treat CF patients. The researchers ultimately concluded that N-of-1 studies could be used effectively to investigate the effects of CFTR mutations in CF patients with varying genotypes and phenotypes [84].

The continued development of novel CF therapies will require better, more innovative methods to test drug efficacy in patients. Large clinical studies are not often feasible, requiring the use of more individualized N-of-1 trials. N-of-1 trials are costly, however, and place the participants at increased risk of having adverse drug effects. Additionally, N-of-1 trials are lengthy, with little ability for scale up and limited drug screening for individual patients. Attrition and adherence may also be a problem. In the San Francisco study, only 7 out of 10 patients completed the study with lack of uniform drug adherence, a known issue in CF treatment that limits assessing therapeutic efficacy [86]. The N-of-1 study approach is promising despite these potential limitations but will not be sufficient to support advancement of CF therapies, even when performed in addition to typical clinical trials. Innovative trial designs and the development of novel in vitro and in vivo animal models is necessary; these resources may support the safer and more effective screening and testing of potentiators and combination drugs to advance personalized CF treatment.

## 5. Animal Models in CF

Animal models remain a vital resource used to study CF pathophysiology and to perform pharmaceutical testing. However, animal studies can be costly, lengthy, and in some cases, controversial [87]. The murine model remains one of the most advantageous because of easy gene manipulation, short gestation times, and relatively low housing costs. However, the mouse model does not fully capture the multi-organ phenotype of CF [75]. While CF pigs display similar phenotypes to humans, pigs have a 100% myocardial infarction risk, leading to increased breeding and maintenance costs. Aside from cost, efficacy is also a concern with the use of animal models. A review by Knight noted that many existing animal experiments are insufficiently predictive of human outcomes and therefore do not provide substantial benefits in the development of human clinical interventions [88]. Further examination of animal models in the development of novel clinical interventions revealed that animal models significantly contributed toward the development of clinical interventions in only 2 out of 20 cases [89]. In studying the human lung, in vitro models are particularly powerful in increasing the understanding of lung physiology, especially in functional regions that are difficult to access, such as in alveolar compartments [90]. Leveraging 3D in vitro models could aid in discovering new therapeutic targets, acting as an important bridge in drug testing between animal models and human clinical trials.

## 6. Organs-on-a-Chip

### 6.1. Microfluidic-Based Organs-on-a-Chip

Cell culture continues to be cost effective and lends itself to high throughput screening while circumventing the ethical and regulatory considerations pertaining to animal models. The development of unique in vitro models provides a system that can not only mimic tissue–tissue interactions but can also mimic the physical microenvironment in which the cells of interest are found. The application of microfluidic-based organ-on-a-chip models to reproduce complex structural and functional living human organs is indispensable in understanding diverse biological responses and elucidating disease mechanisms. Microfluidic device can be fabricated through photolithography and soft lithography techniques (Figure 2A) [55].

### 6.2. Advantages of Organs-on-a-Chip

PDMS has been widely used to fabricate organs-on-a-chip because of its biocompatible material and its biomechanical characteristics [91]. PDMS has high gas permeability allowing cellular oxygenation, an issue that previously required an additional oxygen generator to supply sufficient oxygen to cultured cells [92]. Removing the need for supplemental oxygen allows researchers to culture cells with high metabolic demands, expanding the range of organs that can be developed. Another advantage of PDMS is its optical transparency. Specifically, real-time visualization is possible using high-resolution phase-contrast and fluorescence microscopy imaging [55,87]. In addition, PDMS organ-on-a-chip is easily portable and can be connected to a peristaltic pump to feed cells automatically with various flow rates. The ability to control fluid flow allows for better control of the cellular environment. These organs-on-a-chip allow for the reproduction of both the physiological fluid shear and pulsatile flow patterns that more effectively mimic human organs [87]. Furthermore, the small size of the cell culture chambers minimizes the consumption of reagents, providing opportunities to achieve high throughput and high-resolution analysis [55].

### 6.3. Limitations of Organs-on-a-Chip

One of the challenges using microfluidic-based organ-on-a-chip is preventing air bubbles within the system. The cell culture chamber thickness is on a micron scale, and air bubbles can easily enter and become trapped inside the chamber. Cells in the culture chamber can be washed away or damaged as air bubbles pass. In order to prevent this complication, the temperature of the refreshing media and the media in the chip must be the same, and all surfaces in the chamber must be smooth, including the tubing.

A major issue with regard to PDMS is that small molecules can adsorb onto the highly hydrophobic PDMS surface (contact angle > 75°) (Figure 3A) [91,93]. This is a notable disadvantage in studies focused on drug delivery and drug discovery [94,95]. To address this challenge, the PDMS surface is exposed to oxygen plasma during the fabrication of the organ-on-a-chip. This process changes the surface to become highly hydrophilic (contact angle < 20°) (Figure 3A). Silanol-functional groups (SiOH) are also created on the PMDS surface. Contact between the silanol groups results in the formation of siloxane bridges (Si-O-Si) on the PDMS surface [96]. This allows permanent bonding of the PDMS surface with glass, or another PDMS surface, without requiring an adhesive. The wettability of the plasma treated PDMS surface, however, is not stable. Regression back to a highly hydrophobic surface occurs within three days (Figure 3B). To overcome this limitation, a smart PDMS has been developed by adding PDMS-PEG block copolymer to the normal PDMS before casting the substance into the mold [91]. This smart PDMS maintains a hydrophilic surface for up to 20 months, with reduced non-specific adsorption of IgG, albumin, and lysozyme [91]. Furthermore, PDMS can be maintained as a hydrophilic surface by coating with the hydrophilic polyvinyl alcohol (PVA) [93]. We demonstrated that PVA coated PDMS maintains a hydrophilic surface (contact angle < 20°) for at least 3 months (Figure 3B).

### 6.4. CF Modeling in the Lung

Organ-on-a-chip is also beneficial, as it may be used in personalized human disease models. We successfully monitored the CFTR function of patient-derived lung airway epithelial cells obtained from CF (F508del/F508del) and non-CF patients in a highly sensitive manner (Figure 2B). The short-circuit current assay, the gold standard method to monitor CFTR function, requires at least 100,000 cells. However, using an iodide efflux assay, the microfluidic device required only 10,000 lung airway epithelial cells to monitor CFTR function in response to the cAMP-activating agonist forskolin. The F508del-mutant airway epithelial cells showed a lack of CFTR function. In another study, the microfluidic device was used to assess the effect of IL-13 on asthmatics and the development of COPD [97]. The authors showed that the model was successful in mimicking the function and structure of the human airway and was more robust than previous in vitro models. Furthermore, the device served as a model of pulmonary disease and reproduced the inflammatory lung responses in vitro [97]. The ability of the microfluidic device to mimic in vivo organ-level responses offers a powerful manipulatable in vitro model to study human pathophysiology and preclinical drug evaluation. A report by Huh et al. [87] discussed the development of a microfluidic device with the ability to mimic the distal alveolar-capillary interface of the human lung where gas exchange occurs. The lung-on-a-chip is composed of two cell culture chambers separated by a thin porous membrane that allows the researcher to replicate the important structural, functional, and mechanical properties of the alveolar-capillary interface, producing a novel model for studying the microphysiology of the gas exchange regions of the lung. Human alveolar epithelial cells and human pulmonary microvascular endothelial cells were co-cultured on opposite sides of an extracellular matrix-coated membrane in this study. The co-culture of the cells in a special type of microfluidic device with two separate channels allowed for the use of different reagents, with one channel bathing the epithelium that is in contact with air in the lung and the other giving access to the endothelium lining capillaries that contain blood in the lung. Once cell confluency was achieved, the epithelial side of the channel was exposed to air to produce the air–liquid interface of the lung, and the physiologic action of breathing was emulated by lining both sides of the culture compartments with lateral chambers attached to a vacuum pump. This system allowed for the distortion of the epithelial cells, representing the action of breathing in a living human lung [87]. This model is valuable for physiological studies in the distal gas exchange region of the lung however, additional microfluidic chips are needed to mimic the diverse tissue–tissue interactions present in other functionally distinct compartments of the lung, necessitating the development of additional microfluidic chips to mimic diverse tissue–tissue interactions in the lung.

In contrast to the gas exchange function of the lung alveoli, the conducting airways function to warm, humidify, and clear inhaled air of environmental toxins and infectious microorganisms. A mucociliary escalator that predominates in the larger airways is critical for ridding the lungs of noxious agents, as evidenced by the recurrent infections that plague people with CF and other lung diseases characterized by defects in mucociliary clearance. Thus, distinct epithelial cells, namely mucus secreting goblet cells and ciliated cells, that line the large airways differ significantly from the type I and type II epithelial cells specialized for gas exchange function in the distal alveoli. As the conducting airways continue to branch and become smaller in size, the epithelium progressively transitions from the ciliated and mucous cell rich epithelium to a simple layer of non-ciliated secretory epithelium. The progressive decrease in airway size is also associated with structural tissue changes in the airway walls. Whereas cartilaginous rings provide needed structural support in the trachea to maintain a patent lumen, in smaller conducting airways, the cartilage is replaced by smooth muscle that is critical for regulating airway tone. The diverse tissue–tissue interactions in these functionally distinct regions of the lung necessitate the development of additional microfluidic chips to mimic lung physiology throughout the lung (Figure 4). We propose that the development of four distinct microfluidic chips representing the key spatio-regional and functionally distinct airways from the trachea to the distal alveolar gas exchange region could together provide a surrogate model to accurately represent the pathophysiology along the entire human respiratory tract.

Air enters the oral cavity and travels to the lungs through the trachea, a semi-firm structure with C-shaped cartilaginous rings joined by a fibroelastic ligament and smooth muscle that together protect against airway collapse while allowing for the regulation of lumen size. At this airway level, the surface epithelial cells overlie connective tissue containing fibroblasts with abundant seromucous glands [98,99]. This portion of the lung is not subjected to the intrathoracic pressure changes exerted on the intrapulmonary airways [100,101]. Development of a tracheal model involves two polydimethylsiloxane (PDMS) cell culture chambers stacked on top of one another and separated by a thin porous membrane layer. Human lung epithelial cells are cultured at the air–liquid interface on one side of the permeable membrane. Human seromucous glands are grown on the opposite side with continual perfusion of growth medium. Previous work demonstrated the successful growth and differentiation of both human mucous and serous gland cells [101,102]. The dynamic flow of this system mimics the hemodynamics of the lung microvasculature. This microengineered trachea-on-a-chip can mimic the tissue–tissue interactions in the upper airways of CF patients.

From the trachea, the lung branches into two main bronchi followed by further dichotomous branching subdivisions. While the proximal bronchus resembles the trachea, as more branching occurs, the mural cartilage is lost, with smooth muscle being a prominent component of the airway wall. The changing airway microenvironment necessitates the design of a different model to mimic the human lung bronchus. Again, human epithelial cells are cultured at an air–liquid interface on the porous membrane; both human serous and mucous gland cells and human smooth muscle are cultured on the other side with continual perfusion of growth medium. The inclusion of both serous gland cells and smooth muscle cells mimics the microstructural changes in the lung. More specifically, there is a shift from a more rigid cartilaginous structure in the trachea to the less rigid bronchi where smooth muscle and elastic pressure become more prominent.

As the airways continue to decrease in diameter, cartilage and submucosal glands gradually decrease until these structures no longer exist in smaller airways (bronchioles). The bronchioles mark a distinct change in physiology, as cartilage is no longer present for airway structural support. Consequently, smooth muscle and elastic recoil becomes critical in maintaining airway patency and tone. The microenvironment in the lower airway bronchioles differs from the large airways, with ASM completely encircling the lumen in the airway wall and underlying the airway surface lining epithelium. Replication of this airway region involves seeding airway epithelial cells in the apical compartment with smooth muscle cells seeded in the basolateral compartment.

The conducting airway branches ~23 times before ending in the alveolar sacs where gas exchange occurs [100,101]. The fourth and final regional lung-on-a-chip model mimics cellular interactions critical in gas exchange, namely alveolar type I and II epithelial cells and endothelial cells. As with the previous three models, human lung epithelial cells are cultured at the air–liquid interface of the membrane, with human endothelial cells cultured in the bottom chamber. This tissue–tissue interaction accurately represents the microenvironment of the alveolar-capillary region in the human lung.

Each of these microfluidic devices simulates isolated regions of the human lung. Taken together, the four regio-spatially distinct lung-on-a-chip models provide an in vitro system that accurately mimics tissue–tissue interactions and the physical microenvironment at distinct regions of the airway (Figure 4). Using CF patient-derived cells, this CF-focused model provides opportunities to investigate the effects of diverse CFTR mutations and the resultant altered CFTR protein functions. All of this can be accomplished in an experimental system that may be manipulated and mimics the functionally distinct airway regions.

### 6.5. CF modeling in the Pancreas

The human pancreas is complex organ that plays an important role in maintaining exocrine and endocrine functions. Multiple pancreatic organ-on-a-chip models have been introduced to investigate insulin secretion from rat pancreatic islets [103], patient-derived human islets [104], the human pancreatic β-cell line [105], and human induced pluripotent stem cells (iPSC)-derived islets [106]. However, CFTR is only expressed in the pancreatic ductal epithelial cells [55], and an additional cell culture chamber is critical to investigate cystic fibrosis-related disorder such as CFRD. We previously developed a patient-derived pancreas-on-a-chip with two perpendicular cell culture chambers. We have successfully co-cultured patient-derived pancreatic ductal epithelial cells in the top chamber and pancreatic islets in the bottom chamber to mimic pancreatic physiology. We demonstrated the existence of cell–cell communication between these two cell types (Figure 5) [55]. We observed that defective CFTR function in pancreatic ductal epithelial cells directly reduces insulin production in pancreatic islets co-cultured in the adjacent chamber in the pancreas-on-a-chip [55]. With this chip model, we were able to monitor both CFTR function in patient-derived pancreatic ductal epithelial cells with less than 10,000 cells (Figure 2C) and insulin secretion in pancreatic islets with only 15 islets in a highly sensitive manner (Figure 2D) [55]. In addition, we were able to monitor amylase secretion with only 1000 acinar cells, one of major pancreatic cell types that produces digestive enzymes (Figure 2E). These examples provide evidence that these fabricated organs-on-a-chips are capable of modeling complex human organ-level functions.

### 6.6. CF Modeling in the GI Tract

Human intestinal epithelial cells secrete gel-forming mucin-2 (MUC-2) in the healthy intestine to protect against bacteria [107]. Donald E. Ingber’s group developed a human colon-on-a-chip to investigate human colonic mucus physiology [107]. They successfully differentiated patient-derived colonic epithelial cells into MUC-2 secreting goblet cells mimicking in vivo physiology and monitored mucus secretion depending on 3′,5′-cyclic adenosine monophosphate (cAMP)-mediated ion channel function with CFTR. This model will help us to understand the role of mucus in intestinal homeostasis in CF patients. Furthermore, this in vitro co-culturing model allows researchers to investigate drug toxicity in human organs, such as in the liver and gut [108,109]. In consideration of the in vivo structure where the blood vessels are located in close proximity to the intestinal epithelial cells, we have designed a new gut-on-a-chip model (Figure 6). This system allows for the co-culture of intestinal epithelial cells and vascular endothelial cells in the same chip to investigate drug toxicity-related effects and gut-vascular barrier function [110]. Passage of the gut microbiota into the blood stream is blocked by the gut-vascular barrier. This technology allows researchers to study patient-specific disease and pathophysiologic disease mechanisms to provide rational for the development of personalized therapeutic approaches.

### 6.7. Precision Medicine Using Organs-on-a-Chip

The developed microfluidic device systems may also be applied to purposes other than studying the pathophysiology of CF in the lung. These organs-on-a-chip provide the opportunity to advance targeted medical treatment for CF patients. Many current treatments are prescribed based on the general success rate of a drug in a large population. However, this new approach enables precision medicine with drug selection and prescription based upon the specific response of an individual patient’s cells to a given drug [111]. CF has been proposed as a strong candidate disease for using the organ-on-a-chip concept since patient-specific pharmacotherapy is required for individualized CFTR mutations [89]. Development of a personalized therapeutic strategy using the individual patient’s lung physiology can be achieved by collecting samples directly from the patient and creating individual microfluidic models by culturing the resultant cells. This patient-derived organ-on-a-chip then allows for the selection of specific CFTR pharmacotherapy that is tailored to the individual based on their respective CFTR mutation and their response to the drug. The use of personalized models such as this also have the potential to advance understanding of CF-related disease mechanisms by identifying phenotypic variability to drug response in patients with the same CFTR mutation as well as facilitating the development of cost-effective approaches for selecting efficacious drug regimens for CF patients.

## 7. Conclusions

Individualized pharmacotherapy has the potential to provide a feasible, cost-effective treatment options for CF. Traditional use of animal models requires substantial time and resources and often fails to predict toxicity or efficacy of a drug in humans. The microengineered 3D microfluidic-based organs-on-a-chip mimic the biomechanical and biological microenvironments within human organs. These systems provide a viable approach to better predict drug efficacy and toxicity in individual patients. The organs-on-a-chip devices are also amenable to scale-up for commercial use in pharmaceutical applications. This high throughput system has potential utility in advancing patient care by enhancing efficient screening of new potential CF therapies and supporting new drug discovery.

## Figures and Tables

**Figure 1 micromachines-12-00747-f001:**
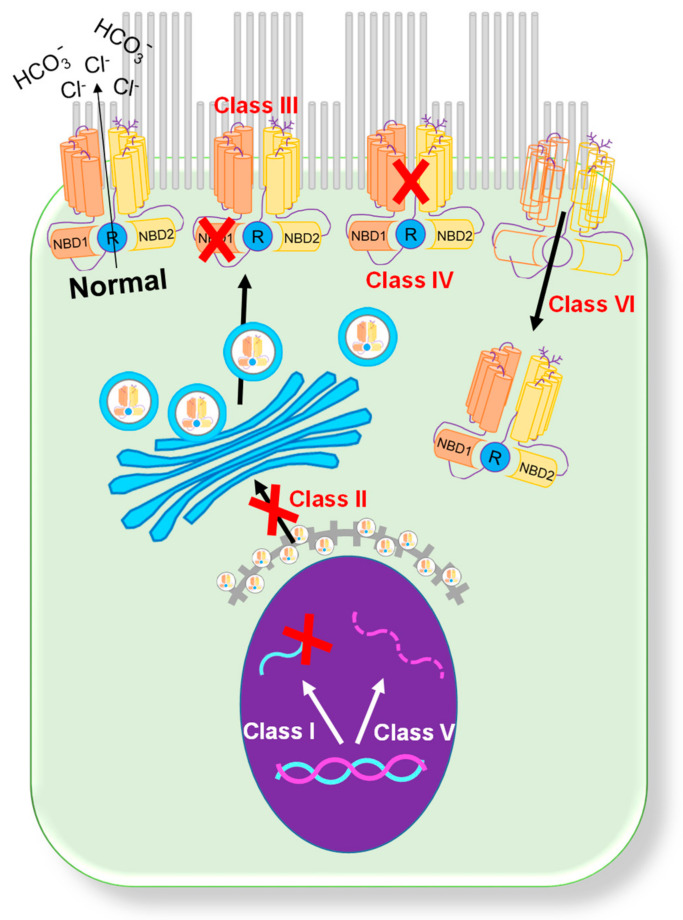
**Classification of CFTR mutations.** A schematic showing six categories of CFTR mutations [4].

**Figure 2 micromachines-12-00747-f002:**
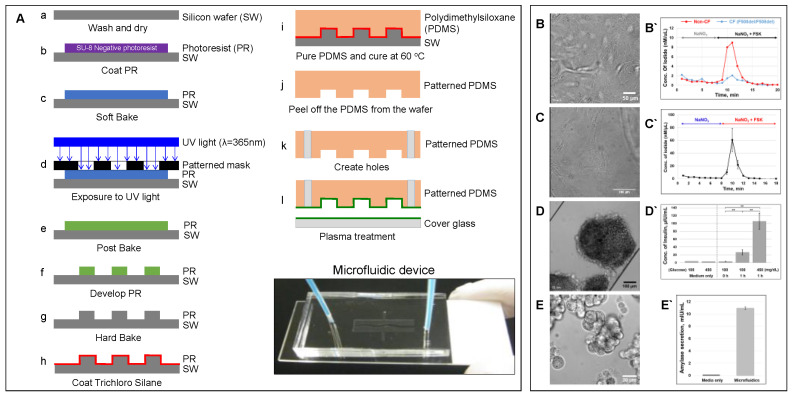
**Microfluidic-based organ-on-a-chip.** (**A**) Schematic showing a protocol of standardized photolithography and soft lithography techniques to fabricate a microfluidic device [55]. Patient-derived lung airway epithelial cells obtained from CF (F508del/F508del) and non-CF donors (**B**), pancreatic ductal epithelial cells (**C**) [55], aggregated pancreatic islets (**D**) [55], and aggregated acinar cells (**E**) were cultured in microfluidic devices and monitored for CFTR function (**B`**–**C`** [55]) using an iodide efflux assay. Insulin (**D`**) [55] and amylase secretion (**E`**) are also shown. To help cell attachment in the channel, the cells were coated with collagen prior to seeding. (ELISA analysis with *p-values*: ** < 0.005; number of samples: *n* = 3; Data represented as mean ± SD).

**Figure 3 micromachines-12-00747-f003:**
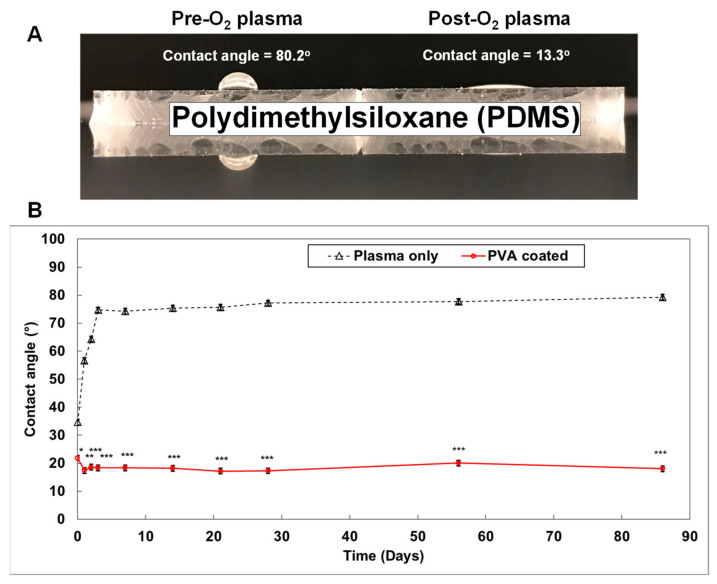
**Generating a smart PDMS by modifying a hydrophobic PDMS surface.** PDMS is a highly hydrophobic material (**A**). When the PDMS surface is exposed to oxygen plasma for 30 s, it becomes highly hydrophilic (contact angle = 13.3°). It returns to a hydrophobic status within 3 days (B; black dotted line). However, the polyvinyl alcohol (PVA) coated PDMS surface maintained a hydrophilic status for over 3 months ((**B**); red solid line). (*p-values*: * <0.05, ** <0.005, *** <0.0005; number of samples: *n* = 3 in each condition; data represented as mean ± SD).

**Figure 4 micromachines-12-00747-f004:**
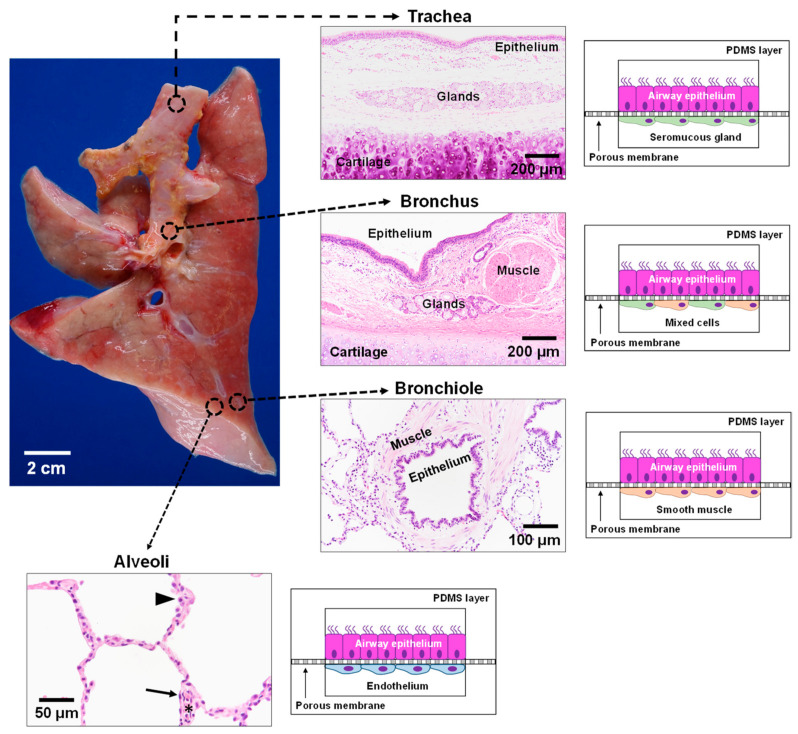
**Schematic of lung-on-a-chip models to represent the unique microenvironments throughout the respiratory tract.** Four separate microfluidic devices represent four functionally distinct regions of the lung. The alveoli are lined by epithelial type II cells (arrowhead) and type I cells (arrow) that are in close proximity to the endothelial cells lining the capillaries containing red blood cells (*). Each lung-on-a-chip model contains different epithelial cell types specialized for region specific functions with distinct cell populations in the subepithelial compartment to mimic the unique cell–cell interactions seen in spatio-regionally distinct lung microenvironments.

**Figure 5 micromachines-12-00747-f005:**
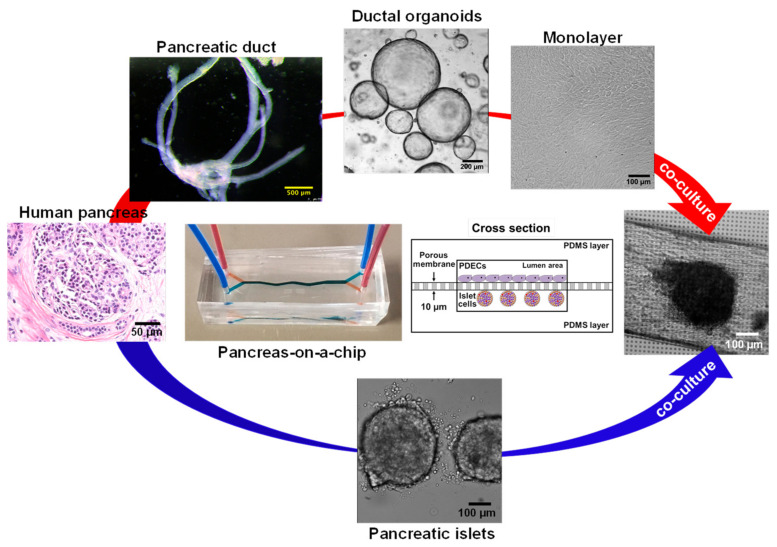
**In vitro co-culturing model, pancreas-on-a-chip** [55]. The pancreas-on-a-chip is composed of two cell culture chambers and a thin layer of porous membrane. Patient-derived pancreatic ductal epithelial cells were plated in the top chamber and polarized on the porous membrane over 4 days. By adding patient-derived islets in the bottom chamber, it mimics the in vivo structure.

**Figure 6 micromachines-12-00747-f006:**
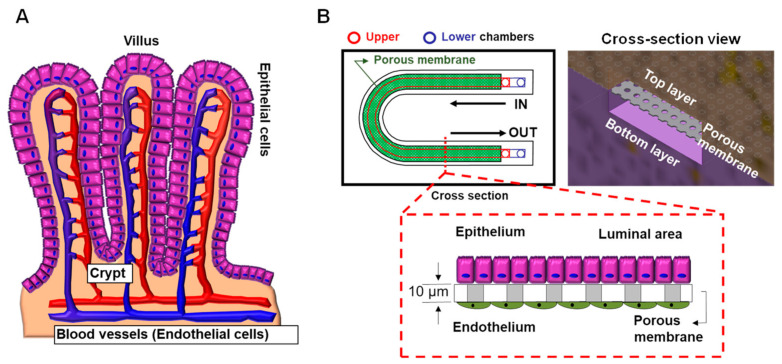
**In vitro co-culturing model, gut-on-a-chip.** (**A**) A schematic representation of human intestinal mucosa with associated vasculature. (**B**) A novel gut-on-a-chip model was designed to mimic the in vivo structure of the human intestinal mucosa. This system allows researchers to co-culture intestinal epithelial cells and endothelial cells in the same chip.

## Data Availability

The data that support the findings of this study are available from the corresponding author upon reasonable request.

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
