# Peer review of "Cystic Fibrosis Human Organs-on-a-Chip"

_micromachines, 2021, doi:10.3390/mi12070747_

Round 1

Reviewer 1 Report

Ogden and co-workers have presented a comprehensive summary of microfluidic organ chips as human experimental models in the research of cystic fibrosis disease.  The review article focuses on an important problem and gives a detailed introduction to the readers.  I therefore recommend it for publication in Micromachines after the following comments are addressed. 

  1. Figures are only used in part 6. I suggest the author add some figures in parts 1 to 5 to make the first part more readable.
  2. Beside the PDMS-based organ-on-a-chip, there are some other types of cystic fibrosis organ chips as well, which deserve a paragraph in part 6.
  3. Biocompatibility is one of the advantages of PDMS chips, and it is also a vital point for cell culture. Discussion about the biocompatibility for such organ-related research is required.

Author Response

1. Figures are only used in part 6. I suggest the author add some figures in parts 1 to 5 to make the first part more readable.

- Response: We thank the reviewer for the suggestion and we added Figure 1 (six categories of CFTR mutations) in part 1 to help the readers more easily understand the CFTR mutation categories. We have also incorporated images and diagrams of CF affected organs (pancreas, gastrointestinal tract and lung) discussed in part 3 into the figures 4, 5 and 6.

2. Beside the PDMS-based organ-on-a-chip, there are some other types of cystic fibrosis organ chips as well, which deserve a paragraph in part 6.

- Response: There are two CF-patient specific organ-on-a-chip models, human pancreas-on-a-chip and human colon-on-a-chip. We have developed pancreas-on-a-chip to study cystic fibrosis related diabetes and observed defective cystic fibrosis transmembrane conductance regulator (CFTR) function in pancreatic ductal epithelial cells affecting reduced insulin secretion from islets. This finding has been published in Nature Communications in 201955. Donald E. Ingber’s group has developed the human colon-on-a-chip and successfully monitored mucus secretion from goblet cells on the chip. This study has been published in Cellular and Molecular Gastroenterology and Hepatology (cmgh) in 2020107. We have added the human colon-on-a-chip in section 6.6. CF modeling in the GI tract.   

(55) Mun, K. S. et al. Patient-derived pancreas-on-a-chip to model cystic fibrosis-related disorders. Nature communications 10, 3124, doi:10.1038/s41467-019-11178-w (2019)

(107) Sontheimer-Phelps, A. et al. Human Colon-on-a-Chip Enables Continuous In Vitro Analysis of Colon Mucus Layer Accumulation and Physiology. Cellular and Molecular Gastroenterology and Hepatology 9, 507-526, doi:https://doi.org/10.1016/j.jcmgh.2019.11.008 (2020).

3. Biocompatibility is one of the advantages of PDMS chips, and it is also a vital point for cell culture. Discussion about the biocompatibility for such organ-related research is required.

- Response: Thank you for bringing to our attention our oversight in not incorporating this critical advantage of PDMS into our review.  The biocompatibility of the flexible and transparent material, PDMS is one of the major advantages key to developing these in vitro cell culture models. We have added the information into section 6.2. Advantages of the Organs-on-a-chip.

Reviewer 2 Report

Overall, the manuscript is well written and potentially fulfills the unmet needs because there has not been a comprehensive review article that covers the modeling of cystic fibrosis using human organ-on-a-chip platforms. Especially, sections 1 through 5 are well-organized and easy to follow. However, in section 6. Organ-on-a-chip, probably the most important section in this article, the authors mainly covered their own works which may not be appropriate considering that this is a review article. Therefore, I highly recommend a major revision to include more comprehensive review on other groups’ work as well as perspectives in modeling CF in other organ systems. Please find more detailed comments below.

Major comments

  1. At the end of section 1, it might be helpful to add a concluding sentence to emphasize that this diverse mutation pattern has been a critical challenge to study CF, and thus, modeling patient-specific disease milieu is really critical.
  2. The quality of the figures is poor. Please use high-resolution images for the final publication.
  3. Because the authors presented three representative CF organs, pancreas, lung, and GI, in section 3, it would be more organized and matched if section 6 can be also categorized (with sub-sections) in the same way. I see that the authors mentioned all three organ platforms in section 6, but it is not catching with the current format. I would recommend having sub-sections something like this. 6.1. microfluidic-based organ-on-a-chip (general introduction to organ chips), 6.2. CF modeling in the lung, 6.3. CF modeling in the GI tract, 6.4. CF modeling in the pancreas (order of 6.2-6.4 can be determined by the authors), 6.5. advantages, 6.6. limitations, 6.7. precision medicine.
  4. When describing how to model CF in the lung, the authors proposed 4 different systems. However, it was not very clear if this approach has been already done somewhere or if this is the authors’ perspectives/plans. It is assumed that this is the authors’ perspective views, but it would be better to clarify. Furthermore, it would be great if the authors can also describe how to model CF in the pancreas and the GI tract. There is almost no discussion with these organ systems.
  5. Figure 2 might benefit from having a schematic or a picture of a complete chip, in addition to showing the lithography and fabrication strategy. The authors mentioned the dimensions of their microfluidic device but the current Figure 2A never dictates that.
  6. A little more descriptions about Figure 2D and 2E may be helpful for better understanding. Do these cell aggregates attach somewhere in the microchannel? Or are they cultured as suspensions in the channel? Embedded in a gel in the channel? Otherwise, having corresponding schematics of each example may be useful.
  7. Not sure if there are published studies, other than the authors’ work, that specifically focused on modeling cystic fibrosis. However, as mentioned above, the authors should cover other groups’ on-chip systems that modeled the pancreas and the GI tract as well. There are various pancreas-on-a-chip and gut-on-a-chip models with different designs. The current manuscript only describes their own work which is not appropriate for a “review” paper. On top of discussing the published work, the authors should suggest how to model cystic fibrosis in the pancreas and the gut.
  8. Discussion on how to reflect patient-specificity is also lacking. Figure 3 shows an approach of using organoids but this could be elaborated more in detail. For instance, organoids generally achieve epithelial cells from clinical samples. However, immune cells, endothelial cells, muscle cells, etc. are still not easy to handle, maintain, and use. The authors can possibly discuss these points.

Minor comment

  1. There is an empty space in line 68, between two sentences. Please check typos.

Author Response

 Major comments

1. At the end of section 1, it might be helpful to add a concluding sentence to emphasize that this diverse mutation pattern has been a critical challenge to study CF, and thus, modeling patient-specific disease milieu is really critical.

- Response: We thank the reviewer for the comment and we have added a concluding statement at the end of section 1. (See below)

“CF-animal models (e.g., mouse11-13, ferret14,15, pig16, rat17, monkey18 and sheep19,20) have been generated to recapitulate human CF mutations, however, it is challenge to reproduce human disease with animal model (e.g., CF-mice have normal lung function)21. To develop novel in vitro CF-model that helps researchers to reproduce human organ physiology to investigate patient-specific disease is critical.”

(11) Tata, F. et al. Cloning the mouse homolog of the human cystic fibrosis transmembrane conductance regulator gene. Genomics 10, 301-307, doi:10.1016/0888-7543(91)90312-3 (1991).

(12) Colledge, W. H. et al. Generation and characterization of a delta F508 cystic fibrosis mouse model. Nat Genet 10, 445-452, doi:10.1038/ng0895-445 (1995).

(13) Zhou, L. et al. Correction of lethal intestinal defect in a mouse model of cystic fibrosis by human CFTR. Science 266, 1705-1708, doi:10.1126/science.7527588 (1994).

(14) Sun, X. et al. Disease phenotype of a ferret CFTR-knockout model of cystic fibrosis. The Journal of clinical investigation 120, 3149-3160, doi:10.1172/JCI43052 (2010).

(15) Fong, S. L., Irimura, T., Landers, R. A. & Bridges, C. D. The carbohydrate of bovine interstitial retinol-binding protein. Prog Clin Biol Res 190, 111-128 (1985).

(16) Ostedgaard, L. S. et al. Processing and function of CFTR-DeltaF508 are species-dependent. Proc Natl Acad Sci U S A 104, 15370-15375, doi:10.1073/pnas.0706974104 (2007).

(17) Trezise, A. E., Szpirer, C. & Buchwald, M. Localization of the gene encoding the cystic fibrosis transmembrane conductance regulator (CFTR) in the rat to chromosome 4 and implications for the evolution of mammalian chromosomes. Genomics 14, 869-874, doi:10.1016/s0888-7543(05)80107-7 (1992).

(18) Dupuit, F. et al. Expression and localization of CFTR in the rhesus monkey surface airway epithelium. Gene Ther 2, 156-163 (1995).

(19) Harris, A. Towards an ovine model of cystic fibrosis. Hum Mol Genet 6, 2191-2194, doi:10.1093/hmg/6.13.2191 (1997).

(20) Fan, Z. et al. A sheep model of cystic fibrosis generated by CRISPR/Cas9 disruption of the CFTR gene. JCI insight 3, doi:10.1172/jci.insight.123529 (2018).

(21) McCarron, A., Donnelley, M. & Parsons, D. Airway disease phenotypes in animal models of cystic fibrosis. Respiratory Research 19, 54, doi:10.1186/s12931-018-0750-y (2018).

2. The quality of the figures is poor. Please use high-resolution images for the final publication.

- Response: Our apologies, we have replaced the poor quality figures with revised figures with high-resolution images and diagrams.

3. Because the authors presented three representative CF organs, pancreas, lung, and GI, in section 3, it would be more organized and matched if section 6 can be also categorized (with sub-sections) in the same way. I see that the authors mentioned all three organ platforms in section 6, but it is not catching with the current format. I would recommend having sub-sections something like this. 6.1. microfluidic-based organ-on-a-chip (general introduction to organ chips), 6.2. CF modeling in the lung, 6.3. CF modeling in the GI tract, 6.4. CF modeling in the pancreas (order of 6.2-6.4 can be determined by the authors), 6.5. advantages, 6.6. limitations, 6.7. precision medicine.

- Response: We thank the reviewer for the suggestion and we have modified the sub-sections in section 6 in the main manuscript as follows;

6.1. microfluidic-based organs-on-a-chip; 6.2. Advantage of organs-on-a-chip; 6.3. Limitations of organs-on-a-chip; 6.4. CF modeling in the lung; 6.5. CF modeling in the pancreas; 6.6. CF modeling in the GI tract; 6.7. Precision medicine.

4. When describing how to model CF in the lung, the authors proposed 4 different systems. However, it was not very clear if this approach has been already done somewhere or if this is the authors’ perspectives/plans. It is assumed that this is the authors’ perspective views, but it would be better to clarify. Furthermore, it would be great if the authors can also describe how to model CF in the pancreas and the GI tract. There is almost no discussion with these organ systems.

- Response: We thank the reviewer for sharing the concerns and comments which have led to revisions that improve the manuscript. An explanation has been added to discuss in more detail the reasoning for us proposing 4 different systems in Section 6.4. CF modeling in the Lung. This will help the reader better understand the rationale for needing spatio-regional functionally distinct lung-on-a-chip models to model the human respiratory tract. In addition, we have also added how organs-on-a-chip can be used to model CF in the pancreas and GI tract. 

5. Figure 2 might benefit from having a schematic or a picture of a complete chip, in addition to showing the lithography and fabrication strategy. The authors mentioned the dimensions of their microfluidic device but the current Figure 2A never dictates that.

- Response: We thank the reviewer for the suggestion and we have added a picture of the microfluidic device in Figure 2A.

6. A little more descriptions about Figure 2D and 2E may be helpful for better understanding. Do these cell aggregates attach somewhere in the microchannel? Or are they cultured as suspensions in the channel? Embedded in a gel in the channel? Otherwise, having corresponding schematics of each example may be useful.

- Response: Pancreatic islets (Islets of Langerhans) are composed of multiple cell types (α-cells which produce glucagon; β-cells which produce insulin; δ-cells which produce somatostatin; pp cells which produce pancreatic polypeptide). We have further described this information in the manuscript (please see Section 3.1. Pancreas). Acinar cells (Figure 2E) comprise the glandular structures at the distal termination of the pancreatic duct lined by epithelial cells. Acinar cells secrete digestive enzymes which is carried by the pancreatic duct to the intestine. We seeded aggregated pancreatic islets or acinar cells into the microfluidic device. To help cell attachment in the channel, collagen was coated prior to seeding the cells. This additional information has been added to the Figure 2 legend.

7. Not sure if there are published studies, other than the authors’ work, that specifically focused on modeling cystic fibrosis. However, as mentioned above, the authors should cover other groups’ on-chip systems that modeled the pancreas and the GI tract as well. There are various pancreas-on-a-chip and gut-on-a-chip models with different designs. The current manuscript only describes their own work which is not appropriate for a “review” paper. On top of discussing the published work, the authors should suggest how to model cystic fibrosis in the pancreas and the gut.

- Response: We thank the reviewer for pointing out the need to make the review more comprehensive. To date, there are two organ-on-a-chip models published regarding cystic fibrosis as described above. (i.e., human pancreas on a chip to study cystic fibrosis related diabetes and human colon-on-a-chip to study mucus secretion). All other pancreatic organ-on-a-chip models have been developed to study insulin secretion from pancreatic islets where CFTR is not expressed. Since this model is focused on non-CF related diabetic research, we chose to include instead the in vitro co-culturing model to investigate cystic fibrosis-related diabetes (CFRD) since the focus of the review is on CF related organ-on-a-chip models. The conclusions using this model are also included, namely that defective CFTR function in pancreatic ductal epithelial cells directly reduced insulin secretion in islets.

8. Discussion on how to reflect patient-specificity is also lacking. Figure 3 shows an approach of using organoids but this could be elaborated more in detail. For instance, organoids generally achieve epithelial cells from clinical samples. However, immune cells, endothelial cells, muscle cells, etc. are still not easy to handle, maintain, and use. The authors can possibly discuss these points.

- Response: To address patient-specificity, we have highlighted the over 2000 CFTR mutations identified along with the mutation classification into six categories to target pharmaceutical therapy. We have also described patient-specific representative CFTR mutations in each category in section 1 and introduced examples of clinical trials with FDA-approved drugs targeted to rescue CFTR function for specific mutations section 2. We have also discussed N-of-1 clinical trials as a means to address patient specificity to drug response. In this review, we have introduced patient-derived organ-on-a-chip models to investigate cystic fibrosis or cystic fibrosis related disorders and focused on epithelial cells wherein CFTR is highly expressed and the cell types in direct vicinity to the CFTR expressing cells that have potential cell-to-cell interactions. Organ-on-a-chip models incorporating these other cell types including lung seromucous cells, lung smooth muscle cells, lung capillary endothelial cells, pancreatic islet cells, and pancreatic acinar cells are discussed and represented in the figures.  

Minor comment

1. There is an empty space in line 68, between two sentences. Please check typos.

- Response: Our apologies, we have removed the empty space in the line and checked for typographical errors in the main manuscript.

Round 2

Reviewer 2 Report

Concerns have been well addressed through the revision.